# Electronic Structure and Optical Properties of Inorganic *Pm3m* and *Pnma* CsPbX_3_ (X = Cl, Br, I) Perovskite: A Theoretical Understanding from Density Functional Theory Calculations

**DOI:** 10.3390/ma16186232

**Published:** 2023-09-15

**Authors:** Hamid M. Ghaithan, Saif M. H. Qaid, Zeyad A. Alahmed, Huda S. Bawazir, Abdullah S. Aldwayyan

**Affiliations:** 1Physics and Astronomy Department, College of Science, King Saud University, Riyadh 11451, Saudi Arabia; sqaid@ksu.edu.sa (S.M.H.Q.); zalahmed@ksu.edu.sa (Z.A.A.); 438204457@student.ksu.edu.sa (H.S.B.); 2King Abdullah Institute for Nanotechnology, King Saud University, Riyadh 11451, Saudi Arabia

**Keywords:** inorganic perovskite, DFT calculation, electronic properties, optical properties

## Abstract

In this study, we investigated the optoelectronic properties of cubic (*Pm3m*) and orthorhombic (*Pnma*) CsPbX_3_ (X = I, Br, and Cl). We utilized the full potential linear augmented plane wave method, which is implemented in the WIEN2k code, to facilitate the investigation. Different exchange potentials were used to analyze the optoelectronic behavior using the available density functional theory methods. Our findings revealed that CsPbX_3_ perovskites display direct band gaps at the R and Г points for cubic (*Pm3m)* and orthorhombic (*Pnma)* structures, respectively. Among the exchange potentials, the mBJ-GGA method provided the most accurate results. These outcomes concurred with the experimental results. In both *Pm3m* and *Pnma* structures, interesting changes were observed when iodide (I) was replaced with bromine (Br) and then chlorine (Cl). The direct band gap at the R and Г points shifted to higher energy levels. Similarly, when I was replaced with Br and Cl, there was a noticeable decrease in the absorption coefficient, dielectric constants, refractive index, and reflectivity, in addition to a band gap shift to higher energy levels.

## 1. Introduction

Organic–inorganic perovskites have become a focal point in research related to photovoltaics, light-emitting diodes, lasers, and photodetectors [1,2,3,4,5,6,7,8,9,10,11,12,13,14,15,16,17,18,19,20]. The chemical formula for an inorganic halide perovskite is ABX_3_, where A represents an inorganic monovalent cation (such as Cs), B signifies an inorganic divalent cation (Pb or Sn), and X denotes a halogen anion (I, Br, or Cl) [21]. Numerous researchers have utilized various density functional theory (DFT) methods to understand the optoelectronic properties of organic–inorganic perovskites [22,23,24,25,26,27,28,29,30,31,32,33,34,35,36,37,38,39,40,41,42,43]. Local density approximation (LDA) and Perdew–Burke–Ernzerhof generalized gradient approximation (PBE-GGA) are the most commonly used methods. These models are preferred because of their low computational cost [37,38,44,45]. However, they struggle to calculate the energy band gap (E_g_) and lattice parameters of the studied perovskite owing to the self-interaction error and the lack of a derivative discontinuity for comparing the Kohn–Sham (KS) band gap with the experimental band gap. The Engel–Vosko (EV-GGA) functional, on the other hand, is designed to reproduce the exchange correlation potential rather than the total energy and is useful for calculating band gaps and optical properties while overestimating lattice parameters [35,36,37,38,39,40,41]. An alternative method, the (screen) hybrid functional, offers superior accuracy in terms of E_g_ but at a higher computational cost [46,47].

Previous research has demonstrated that the modified Becke–Johnson GGA (mBJ-GGA) yields accurate E_g_ values for a wide range of materials, such as wide-band-gap insulators, semiconductors, and 3d transition-metal oxides. This is due to its additional dependence on kinetic energy density [25,42,43,48,49,50,51,52,53,54,55]. The mBJ-GGA method has been applied to study the optoelectronic properties of cubic (*Pm3m)*, tetragonal (*P4mm)*, and orthorhombic *Pnma* CsPbBr_3_ perovskite. The calculated E_g_ aligns well with the experimental values. Moreover, the modified Becke–Johnson generalized gradient approximation (mBJ-GGA) potential can be used to study the optoelectronic properties of *Pm3m* and *Pnma* CsPbX_3_. The systematic DFT investigation of *Pm3m* and *Pnma* CsPbX_3_ perovskites provides a deep understanding of their crystal structure–property relationships, as well as their potential applications in optoelectronics.

In our study, we investigated the optoelectronic properties of *Pm3m* and *Pnma* CsPbX_3_ (X = I, Br, and Cl) using the accurate mBJ-GGA method, both with and without spin–orbit coupling (SOC). Along with LDA [35], the PBE-GGA [37], Engel–Vosko GGA (EV-GGA) [56], new modified Becke–Johnson GGA (nmBJ-GGA), and unmodified Becke–Johnson GGA (umBJ-GGA) methods were used. These methods were implemented in the WIEN2k code. To validate the DFT calculation, the E_g_ values were compared with previous experimental and theoretical results. The mBJ-GGA method yielded values that aligned well with the experimental data. This study endeavors to delve into the cubic and orthorhombic crystal systems with a renewed perspective, taking advantage of recent advancements in characterization techniques and computational modeling. The calculated electronic and optical properties show tunable absorption coefficients, dielectric constants, refractive indices, and reflectivity values, as well as charge transport properties, when iodide is replaced with bromine and chlorine within the visible light range. As a result, our findings are critical for furthering research into CsPbX_3_ perovskite materials’ potential applications in optoelectronic devices such as solar cells, light-emitting diodes, and photodetectors.

## 2. DFT Methods

In this study, we employed the full potential linear augmented plane wave (FP-LAPW) method within the framework of DFT. This method is incorporated in the WIEN2k code [30,57,58,59] as outlined in our previously published research [42,54,60,61,62]. Our primary focus was to investigate the optoelectronic properties of *Pm3m* and *Pnma* CsPbX_3_ (X = I, Br, and Cl). Therefore, we used the following methods: LDA [44], PBE-GGA [37], mBJ-GGA [48], nmBJ-GGA, umBJ-GGA, and EV-GGA, as shown in Figure 1. The PBEsol method was used to investigate the lattice parameters of the structures. This method more accurately reproduces experimental values, as evidenced by its low relative error values [36]. Considering the heavy lead element present, we included the SOC effect [42,63,64] in the calculation with the mBJ-GGA method. The basis function was expanded to R_mt_ × K_max_ = 9 for all structures. We sampled the Brillouin zones using 12 × 12 × 12 and 14 × 9 × 14 k-point meshes for *Pm3m* and *Pnma* CsPbX_3_, respectively. The total energy converged until it reached <10^−4^ Ry. Additionally, we introduced the Fourier expansion of the charge density with a maximum of G_max_ = 12 (a.u.)^−1^. Ultimately, we set the RMT of Cs, Pb, I, Br, and Cl atoms at 2.2, 2.5, 2.5, 2.2 and 2.2 a.u., respectively.

## 3. Results and Discussion

### 3.1. Structural Properties

The structural properties of *Pm3m* and *Pnma* CsPbX_3_ (X = I, Br, and Cl) perovskites were calculated using the PBEsol approximation method. The lattice parameters of these structures were determined by fitting the Murnaghan equation [65]. Figure 2 shows the variation of energy E(Ry) versus the volume of the studied structures, as per the Murnaghan equation of state [65]. For comparison, Table 1 displays the calculated lattice parameters (a, b, and c), pressure derivatives (B′), and bulk modulus values (B), alongside previously measured and predicted values. For cubic CsPbI_3_, CsPbBr_3_, and CsPbCl_3_, we found the lattice parameters to be 6.262, 5.875, and 5.734 A^0^, respectively. For orthorhombic CsPbI_3_, CsPbBr_3_, and CsPbCl_3_, the calculated lattice parameters are (a = 8.856, b = 8.576, c = 12.472 A^0^), (a = 8.161, b = 11.617, c = 8.115 A^0^), and (a = 7.902, b = 11.248, c = 7.899 A^0^), respectively. Our calculated data align well with previous measurements and predictions, thus reinforcing the reliability of our computational scheme [22,29,30,33,61,62,66,67,68,69,70].

Visualization for Electronic and Structural Analysis (VESTA 3) was used to obtain the theoretical X-ray diffraction (XRD) [82], as illustrated in Figure 3. When substituting iodide with bromine, and then with chlorine, the diffraction peaks of *Pm3m* and *Pnma* CsPbX_3_ shifted toward higher 2-theta values. Figure 3a shows the XRD pattern of CsPbI_3_ (*Pm3m*) with diffraction peaks at 13.89°, 19.62°, 23.97°, 31.33°, 34.19°, 39.79°, 42.37°, 44.69°, 46.99°, and 49.32°. These peaks shifted to 15.80°, 22.34°, 27.52°, 31.87°, 35.84°, 39.40°, 45.77°, 48.69°, 51.64°, and 56.88° for CsPbCl_3_ (*Pm3m*). The peaks for orthorhombic CsPbX_3_ (*Pnma*) shifted to higher 2-theta values, as depicted in Figure 3b. This shift was also evident when I was replaced with Br and Cl. This shift is caused by a decrease in the unit cell volume of the crystal structure.

### 3.2. Electronic Properties

The electronic properties of CsPbX_3_ perovskites can vary depending on their crystal structure [1,2,3,4,5,6,7,8,9,10]. The bandgap (E_g_) can be tuned by adjusting the halide component. With increasing halogen atomic number (i.e., Cl < Br < I), the E_g_ of *Pm3m* and *Pnma* decreases. This tunability enables the optimization of solar cell absorption and LED light emission. *Pnma* CsPbX_3_ perovskite has a more complex crystal structure than cubic perovskite, and its electronic properties can differ due to structural changes [22,23,24,25,26,27,28,29,30,31,32,33,34,35,36,37,38,39,40,41,42,43]. Figure 4a shows the band structure along the (X, R, and Γ) K-points of *Pm3m* CsPbX_3_ (X = I, Br, and Cl) using LDA, PBE-GGA, mBJ-GGA, nmBJ-GGA, umBJ-GGA, and EV-GGA, where the top of the valence band was set as 0 eV. The band structure indicated that the valence band maximum (VBM) and the conduction band minimum (CBM) were located at point R structures, resulting in a direct band gap. We also calculated the *Pnma* CsPbX_3_ band structure using the same methods as for CsPbBr_3_, where the VBM and CBM were located at the Γ point with a direct band gap as shown in Figure 4b. As shown in Table 2 and Figure 5, the mBJ-GGA method proved effective at yielding accurate E_g_ values for both *Pm3m* and *Pnma* CsPbX_3_ perovskite. Given the presence of the heavy lead element in CsPbX_3_, it was crucial to include the SOC effect to accurately describe the band structures [60,63,64]. The SOC effect caused a drastic change in the electronic band gap, resulting in band gaps of 0.952, 1.53, and 1.69 eV for *Pm3m* CsPbI_3_, CsPbBr_3_, and CsPbCl_3_, respectively. The band gaps with the SOC effect for *Pnma* CsPbI_3_, CsPbBr_3_, and CsPbCl_3_ were reduced to 1.189, 1.482, and 2.167 eV, respectively. 

The total density of states (TDOS) of both *Pm3m* and *Pnma* CsPbX_3_ was examined to gain insights into the factors influencing E_g_ trends. We began by calculating the TDOS of CsPbX_3_ compounds to observe the effect of replacing I with Br and Cl on the E_g_ trend, as shown in Figure 6. Despite these substitutions, the overall TDOS feature remained consistent, with the DOS edges shifting upward for both the *Pm3m* and *Pnma* CsPbX_3_ compounds. Figure 7 shows the TDOS of the investigated compounds using the mBJ-GGA method, which is known for its accuracy. Furthermore, an upward shift in TDOS was observed for both the *Pm3m* and *Pnma* CsPbX_3_ compounds.

To provide a more detailed analysis, we calculated the partial density of states (PDOS) for *Pm3m* CsPbI_3_ as an example. As shown in Figure 8, Cs ^+^ does not influence the VBM but instead contributes to maintaining overall load neutrality and structural stability [22,26,28,42,63,102,103,104]. The VBM primarily stems from I p orbitals, supplemented by contributions from Pb s orbitals. The CBM is formed primarily by Pb p states, with minor contributions from I s and p states.

### 3.3. Optical Properties

The optical properties of perovskites, specifically *Pm3m* and *Pnma* CsPbX_3_, play a crucial role in determining their performance in solar cells, light-emitting diodes (LEDs), and photodetector devices. The dielectric functions ε_1_(ω) and ε_2_(ω), which describe a material’s response to incident photons as a function of energy, are depicted in Figure 9. The static frequency ε_1_(0) represents the real part of the dielectric function ε_1_(ω) value at zero frequency, and it ranges between 3.02 and 4.60. The radiation absorbed by the compound [105] is represented by ε_2_(ω), with main peaks appearing between 3.51 and 5.50 eV. Notably, ε_2_(ω) remains at zero until photon energy reaches the band gap energy, indicating the onset of direct optical transition between the VBM and the CBM.

Figure 10 depicts the calculated absorption coefficient *α*(*ω*) using the mBJ-GGA method, which determines a light absorber’s ability to harvest solar energy. Moreover, *α*(*ω*) is plotted against energy (0–10 eV) for the *Pm3m* and *Pnma* CsPbX_3_ compounds. The absorption edge shifted upward to a higher energy side when I was replaced with Br and Cl for both *Pm3m* and *Pnma* CsPbX_3_. A complex absorption and dielectric spectrum with multiple peaks can result from the combination of excitonic effects, band-to-band transitions, quantum confinement, phonon modes, defects, and anharmonic effects in CsPbX_3_ perovskite materials. The specific energies and intensities of these peaks can provide important information about the material’s electronic and structural properties, which are important for understanding its behavior and optimizing its performance in various applications. These perovskites exhibit strong light absorption across a broad range of energy, including the visible and near-infrared regions. This high absorption coefficient is attributed to the strong interaction between the lead (Pb) and halogen (X) atoms, which leads to efficient absorption of photons [24,25,26,27,28,29,30,31,32,33,34]. The broad absorption range of these compounds indicates their potential applications in various optical and optoelectronic devices operating within this range.

The refractive index n(ω) was calculated using the mBJ-GGA method as shown in Figure 11. n(*ω*) is a critical feature of semiconductors, indicating the degree to which light is refracted or bent [105]. As the energy increases, the value of n(ω) increases until it reaches 2.74, 2.34, and 2.14 eV for *Pm3m*, CsPbI_3_, CsPbBr_3_, and CsPbCl_3_, respectively, and 2.72, 2.40, and 2.14 eV for *Pnma* CsPbI_3_, CsPbBr_3_, and CsPbCl_3_, respectively. Beyond this point, it starts to fluctuate, exhibiting nonlinear behavior. The calculated n(0) values were 2.138, 1.88, and 1.74 eV for *Pm3m* CsPbI_3_, CsPbBr_3_, and CsPbCl_3_ and 2.14, 1.96, and 1.76 eV for *Pnma* CsPbI_3_, CsPbBr_3_, and CsPbCl_3_, respectively.

The calculated reflectivity R(ω) in relation to incident energy is shown in Figure 12. At zero frequency, *Pm3m* CsPbI_3_ has a static reflectivity R(0) value of 0.132, which then decreases to 0.096 and 0.073 for *Pm3m* CsPbBr_3_ and CsPbCl_3_, respectively. Similarly, the R(0) value of *Pnma* CsPbI_3_ decreases from 0.132 to 0.106 and 0.077 for *Pnma* CsPbBr_3_ and CsPbCl_3_, respectively. By increasing the energy, R(ω) starts to increase in the 3–5 eV range before beginning to fluctuate and decrease at higher energies. The optical properties of *Pnma* CsPbX_3_ perovskites are clearly similar to those of cubic perovskites. They also have tunable optical bandgaps due to their halide composition [24,25]. 

## 4. Conclusions

In summary, this study investigated the optoelectronic properties of *Pm3m* and *Pnma* CsPbX_3_ (X = I, Br, and Cl) perovskite, using different approximation methods in DFT. We used the PBEsol method to calculate the lattice parameters, which were found to align well with the measured and previously predicted values, demonstrating the reliability of our computational scheme. We also calculated the electronic band structure and optical properties of CsPbX_3_ perovskite using different density functional theory methods. The band gap values obtained using the mBJ-GGA method were closely aligned with experimental values. Both *Pm3m* and *Pnma* CsPbX_3_ presented direct band gaps at the R and Г points. When I was replaced with Br and then Cl in *Pm3m* and *Pnma* CsPbX_3_, the direct band gap located at the R and Г points shifted to higher energy levels. This replacement also resulted in a decrease in the absorption coefficient, dielectric constant, refractive index, and reflectivity, along with a band gap shift to higher energy.

## Figures and Tables

**Figure 1 materials-16-06232-f001:**
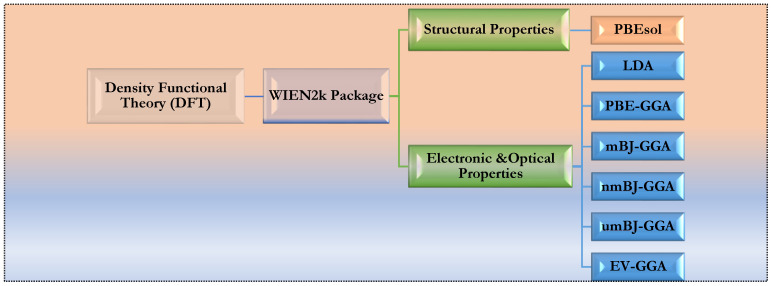
DFT approximation methods used to calculate the optoelectronic properties of *Pm3m* and *Pnma* perovskite structures.

**Figure 2 materials-16-06232-f002:**
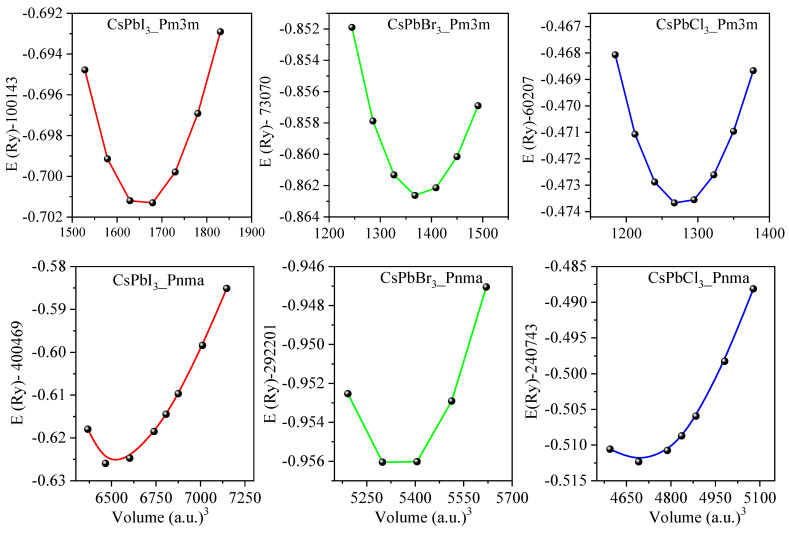
Variation of the energy E(Ry) versus the volume for *Pm3m* and *Pnma* CsPbX_3_(X = I, Br, and Cl) using the PBEsol method.

**Figure 3 materials-16-06232-f003:**
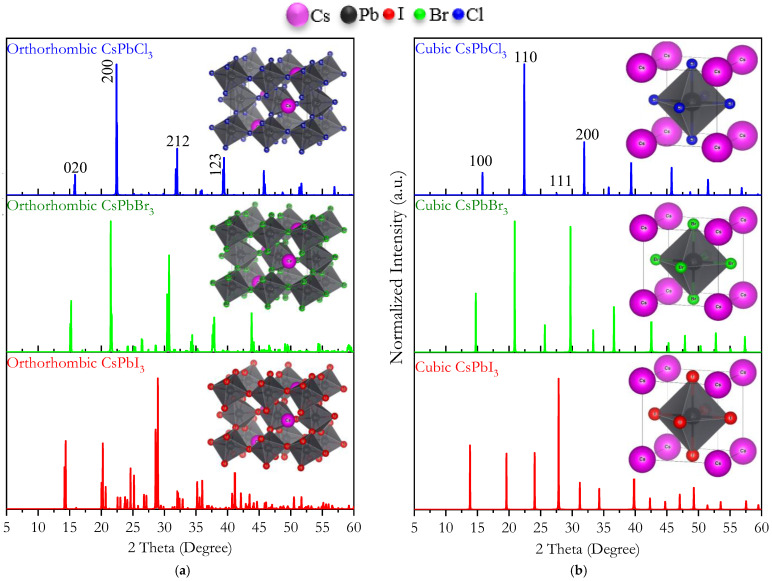
X-ray diffraction patterns for (**a**) cubic and (**b**) orthorhombic CsPbX_3_ (X = I, Br, and Cl) perovskites. Inset: crystal structures of perovskites.

**Figure 4 materials-16-06232-f004:**
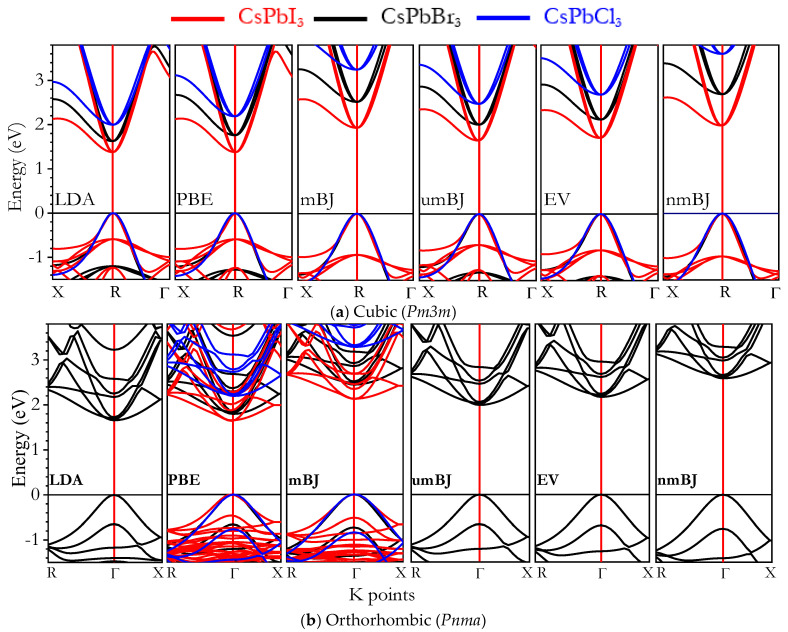
Band structure of *Pm3m* and *Pnma* CsPbX_3_ (X = I, Br, and Cl) perovskites obtained using LDA, PBE, mBJ, umBJ, and nmBJ potentials.

**Figure 5 materials-16-06232-f005:**
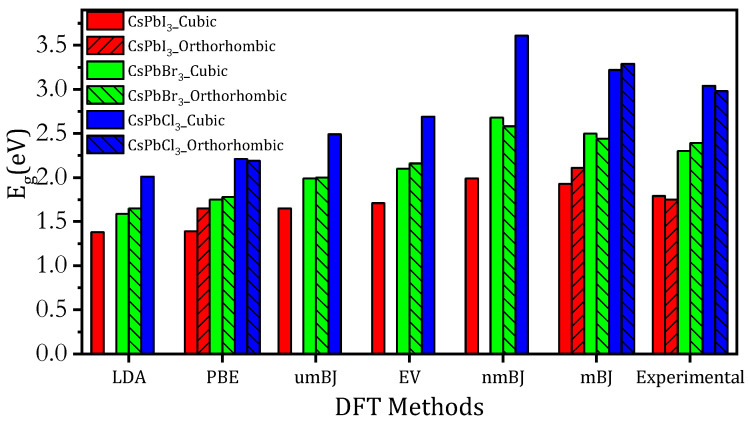
Band gaps (E_g_) of cubic and orthorhombic CsPbX_3_ calculated using various functional methods and compared to experimental E_g_ values.

**Figure 6 materials-16-06232-f006:**
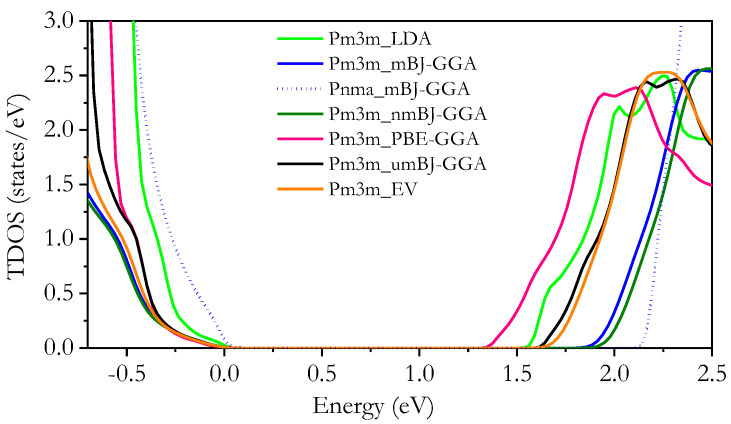
Total density of states of *Pm3m and Pnma* CsPbX_3_ perovskite in the range −0.7 to 2.5 eV using the LDA, PBE-GGA, mBJ-GGA, nmBJ-GGA, umBJ-GGA, and EV-GGA methods.

**Figure 7 materials-16-06232-f007:**
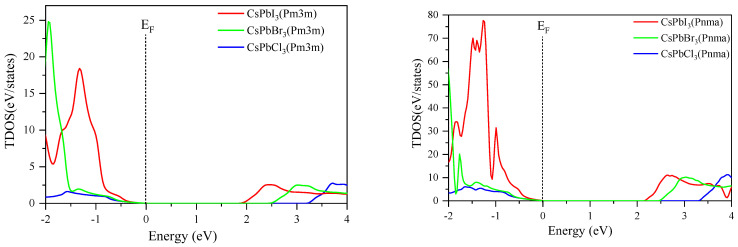
Total density of states of *Pm3m* (**left**) and *Pnma* (**right**) CsPbX_3_ perovskite in the range −0.7 to 2.5 eV using the mBJ-GGA method.

**Figure 8 materials-16-06232-f008:**
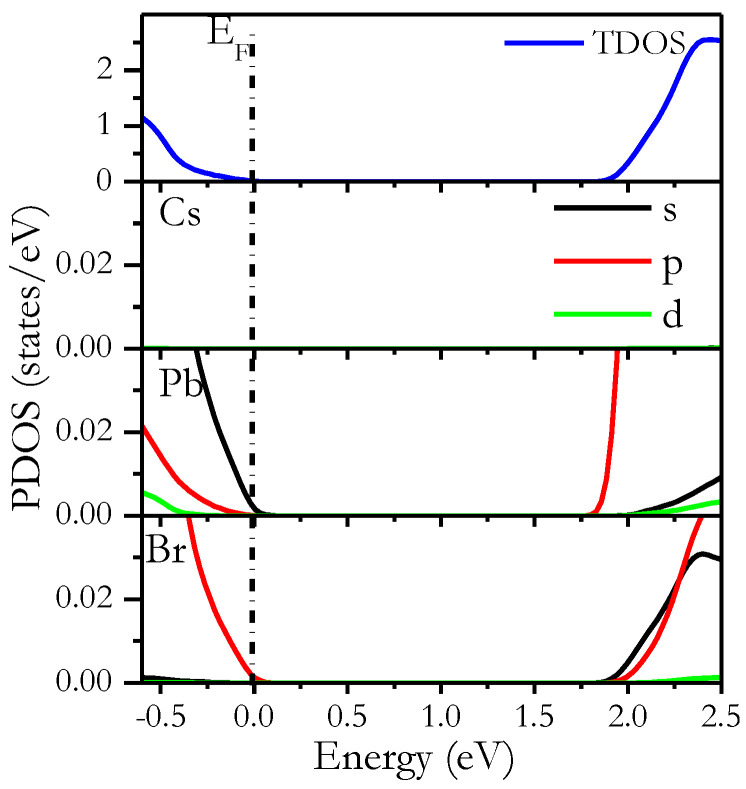
Calculated PDOS of *Pm3m* CsPbI_3_ with various doping concentrations (x) using the mBJ-GGA method.

**Figure 9 materials-16-06232-f009:**
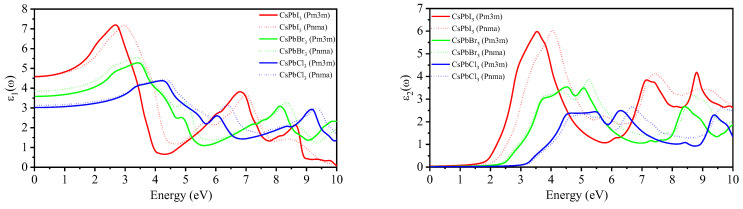
Calculated real dielectric function (**left**) and imaginary dielectric function (**right**) of *Pm3m* and *Pnma* CsPbX_3_ using the mBJ-GGA method.

**Figure 10 materials-16-06232-f010:**
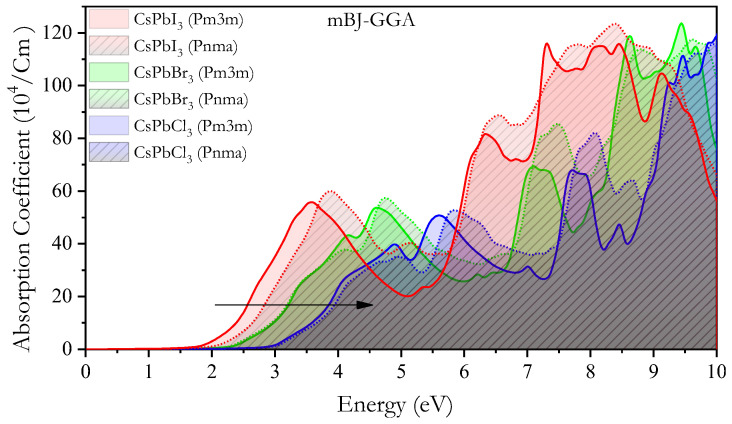
Absorption coefficient α(ω) of *Pm3m* and *Pnma* CsPbX_3_ using the mBJ-GGA method.

**Figure 11 materials-16-06232-f011:**
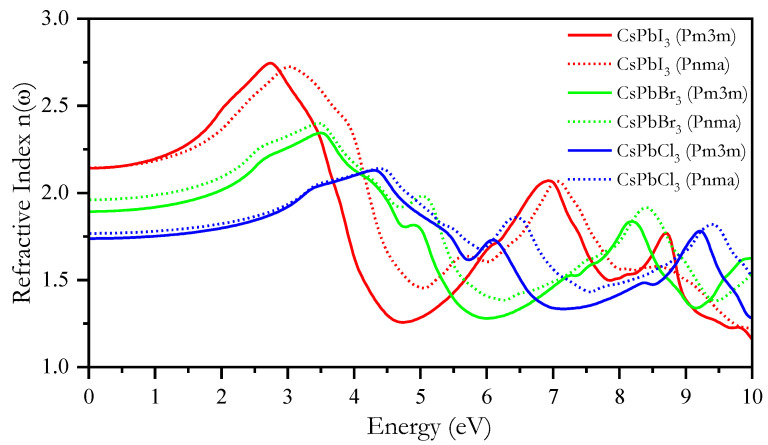
Calculated refraction index values of *Pm3m* and *Pnma* CsPbX_3_ using the mBJ-GGA method.

**Figure 12 materials-16-06232-f012:**
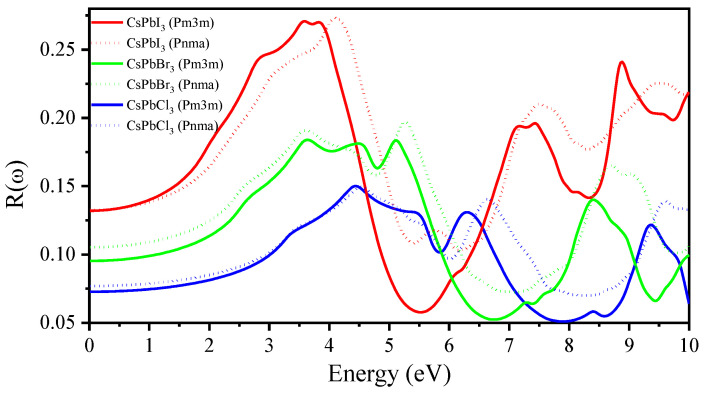
Calculated reflectivity of Pm3m and Pnma CsPbX_3_ using the mBJ-GGA method.

**Table 1 materials-16-06232-t001:** Lattice parameters (a, b, and c), pressure derivatives (B′), and bulk modulus values (B, GPa) for the studied halide perovskites using the PBEsol method.

Lattice Parameters	CsPbI_3_		CsPbBr_3_		CsPbCl_3_	
	*Pm3m* a = b = c	*Pnma*	*Pm3m* a = b = c	*Pnma*	*Pm3m* a = b = c	*Pnma*
This study	6.262	a = 8.856 b = 8.576 c = 12.472	5.875 5.740 [71]	a = 8.161 b = 11.617 c = 8.115	5.734	a = 7.902 b = 11.248 c = 7.899
Other DFT	6.25 [33] 6.39 [29] 6.38 [30] 6.05 [23] 6.40 [33,72]	a = 8.87 [31] b = 8.54 [31] c = 12.45 [31]	5.8445 [73] 5.874 [74]	a = 8.376 [67] b = 11.497 [67] c = 7.617 [67] a = 8.251 [66] b = 11.753 [66] c = 8.203 [66] a = 8.250 [68] b = 11.70 [68] c = 8.210 [68]	5.61 [70] 5.73 [24] 5.49 [23] 5.605 [75,76]	7.973 [61] 11.355 [61] 7.916 [61]
Other exp.	6.177 [77] 6.28 [78] 6.33 [30] 6.297 [79]	a = 8.856 [80] b = 8.576 [80] c = 12.472 [80] a = 8.646 [81] b = 8.818 [81] c = 12.520 [81]		a = 8.260 [73] b = 11.765 [73] c = 8.212 [73]		7.90193 [66] 11.24778 [66] 7.89928 [66] 7.97613 [61] 11.35674 [61] 7.91729 [61]
B (GPa)		17.85	20.74 20.73 [62]	22.65	24.21 22.59 [69] 25.447 [22] 26.33 [70]	25.62
B′		4.357	4.88 4.9 [62]	5.591	5.01 4.33 [69] 4.4 [22]	5.654

**Table 2 materials-16-06232-t002:** Band gaps (E_g_) of Pm3m and Pnma inorganic perovskites calculated using the LDA, PBE-GGA, EV-GGA, PBEsol-GGA, and umBJ-GGA methods and compared to those from previous experimental and theoretical studies.

DFT Approximation Method	CsPbI_3_		CsPbBr_3_		CsPbCl_3_	
	*Pm3m*	*Pnma*	*Pm3m*	*Pnma*	*Pm3m*	*Pnma*
LDA	1.38	--	1.59	1.65	2.01	--
PBE	1.39 0.207^mBJ+ SOC^	1.65	1.75	1.78	2.21	2.19
umBJ-GGA	1.65	--	1.99	2.00	2.49	--
EV-GGA	1.71	--	2.10	2.16	2.69	--
nmBJ-GGA	1.99	--	2.68	2.58	3.61	--
mBJ-GGA	1.93 0.952^mBJ + SOC^	2.11 1.189^mBJ + SOC^	2.43 1.53^mBJ + SOC^	2.44 1.482^mBJ + SOC^	3.22 1.69^mBJ + SOC^	3.29 2.167^mBJ + SOC^
Others_DFT	1.45_PBE [54] 1.90_mBJ [54] 1.56 [33] 1.48 [83] 1.11 [23] 1.359 [30] 1.831 [31,84]	1.983_mBJ [60] 1.831 [31,84] 1.48 [83]	1.77_PBE [54] 2.50_mBJ [54] 1.79 [26] 2.00 [85] 1.75 [25] 2.35 [25] 1.12 [23] 1.76 [30] 2.63 [30]	2.32 [86] 2.40 [84] 2.420_mBJ [60]	2.20_PBE [26,87] 2.829_KTB-mBJ [28] 2.92_HSE [24] 3.406_PBE [22] 2.88_GW [71]	3.05 [29] 3.325 [60]
Others_Exp.	1.87 [88] 1.792 [89] 1.85 [61] 1.75 [68]	1.75 [68] 1.85 [61]	2.30 [90] 2.36 [20] 2.32 [91] 2.282 [92]	2.38 [93] 2.479 [61] 2.25 [94] 2.252 [95] 2.36 [96] 2.24 [96] 2.20 [97] 2.32 [98]	3.00 [90] 2.97 [99] 3.04 [100] 2.98 [96]	2.91 [66] 2.78 [101] 3.132 [61]

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
