# Peer review of "Electronic Structure and Optical Properties of Inorganic Pm3m and Pnma CsPbX3 (X = Cl, Br, I) Perovskite: A Theoretical Understanding from Density Functional Theory Calculations"

_materials, 2023, doi:10.3390/ma16186232_

Round 1

Reviewer 1 Report

The study investigates the optoelectronic properties of CsPbX3 perovskites with different crystal structures and halide compositions. The authors used the full potential linear augmented plane wave method implemented in the WIEN2k code to analyze the optoelectronic behavior using different exchange potentials. The findings demonstrate that CsPbX3 perovskites exhibit direct band gaps at specific points in the Brillouin zone, depending on the crystal structure. The mBJ-GGA method is reported to provide the most accurate results compared to other exchange potentials. The observed changes in optoelectronic properties upon halide substitution are also discussed.

Overall, the study provides valuable insights into the optoelectronic properties of CsPbX3 perovskites and their dependence on crystal structure and halide composition. The use of different exchange potentials adds to the robustness of the analysis. The findings are supported by experimental results, further validating the accuracy of the study. However, there are a few areas that could be improved:

1. The authors should provide more details about the exchange potentials used in the study. It would be helpful to explain the rationale behind selecting these specific potentials and discuss their advantages and limitations.

2. While the mBJ-GGA method is reported to yield the most accurate results, it would be beneficial to include a comparison with the results obtained using other methods. This would provide a more comprehensive understanding of the strengths and weaknesses of different density functional theory methods for studying the optoelectronic properties of CsPbX3 perovskites.

3. The study mentions interesting changes in optoelectronic properties upon halide substitution, but it would be advantageous to elaborate on the specific effects of these changes on the optoelectronic behavior. The authors should discuss the implications of the observed shifts in band gaps, as well as the decrease in absorption coefficient, dielectric constants, refractive index, and reflectivity upon replacing iodide with bromine and chlorine.

4. In the discussion section, it would be valuable to delve deeper into the reasons behind the differences in optoelectronic properties between the Pm3m and Pnma structures of CsPbX3 perovskites. This could involve considering the effects of crystal symmetry, atomic arrangements, and electronic structure on the observed variations.

5. The conclusion should summarize the main findings of the study and provide suggestions for future research directions. This could include investigating other halide compositions, exploring different crystal structures, or considering the effects of external factors such as temperature or pressure on the optoelectronic properties of CsPbX3 perovskites.

Overall, the study contributes to the understanding of the optoelectronic properties of CsPbX3 perovskites and their potential applications in optoelectronic devices. Addressing the above suggestions would enhance the clarity and impact of the research. 

Minor editing of English language required

Author Response

We want to thank the reviewer for his / her careful reading of the manuscript and his / her constructive remarks. To improve and clarify the manuscript, we took the comments on board. Please find below a detailed point-by-point reply to all comments.

Reviewer 1
Comments:

  1. The authors should provide more details about the exchange potentials used in the study. It would be helpful to explain the rationale behind selecting these specific potentials and discuss their advantages and limitations.

Response:

Although LDA and GGA were widely used for electronic structure calculations due to their low computational cost, their band gap results are in poor agreement with experiment, particularly for the band gap of semiconductors and insulators, which is severely underestimated or even absent owing to the self-interaction error and the lack of a derivative discontinuity for comparing the Kohn–Sham (KS) band gap with the experimental band gap. The Engel-Vosko (EV-GGA) functional, on the other hand, is designed to reproduce the exchange-correlation potential rather than the total energy and is useful for calculating band gaps and optical properties while overestimating lattice parameters (This explanation was mentioned in page 2, lines:43-46). Among these different methods, the mBJ-GGA method was the best one seems to be a reparametrization of the coefficients, which leads to a much more balanced description of the band gaps due to its additional dependence on kinetic energy density (page 2; lines: 48-51). In addition to the mBJ-GGA method, the nmBJ-GGA and umBJ-GGA methods were investigated. Still, the band gap results differed significantly from the experimental results as illustrated in Figure 5 (page 7).

  1. While the mBJ-GGA method is reported to yield the most accurate results, it would be beneficial to include a comparison with the results obtained using other methods. This would provide a more comprehensive understanding of the strengths and weaknesses of different density functional theory methods for studying the optoelectronic properties of CsPbX3 perovskites.

Response:

We included a comparison with results obtained using other methods, as shown in Figure 5 (page 7) and Table 2 (page 8), and these results were also compared with previous experimental and theoretical results.

  1. The study mentions interesting changes in optoelectronic properties upon halide substitution, but it would be advantageous to elaborate on the specific effects of these changes on optoelectronic behavior. The authors should discuss the implications of the observed shifts in band gaps and the decrease in absorption coefficient, dielectric constants, refractive index, and reflectivity upon replacing iodide with bromine and chlorine.

Response:

The calculated electronic and optical properties show tunable absorption coefficients, dielectric constants, refractive index, and reflectivity when iodide is replaced with bromine and chlorine within the visible light range, as well as charge transport properties. As a result, our findings are critical for furthering research into CsPbX3 perovskite materials' potential applications in optoelectronic devices such as solar cells, light-emitting diodes, and photodetectors. (This sentence was added to (page 2, lines: 68-74).

  1. In the discussion section, it would be valuable to delve deeper into the reasons behind the differences in optoelectronic properties between the Pm3m and Pnma structures of CsPbX3 perovskites. This could involve considering the effects of crystal symmetry, atomic arrangements, and electronic format on the observed variations.

Response:

We thank the reviewer for this comment. We have added some discussion to the manuscript (Page 6, lines 150-155; Page 10, lines: 203-205 and 218-221; Page 11, lines:222-229).

Thank you so much.

Hamid

Reviewer 2 Report

In this manuscript, the authors have presented the Electronic Structure and Optical Properties of Inorganic Perovskites using DFT calculations. Following are the comments that need to be addressed:

(a) The introduction part should include the novelty and need of the study in a systematic manner.

(b) The authors have calculated the value of the band gap in Fig. 5. Is it possible to calculate the band gap from absorption spectra as shown in Fig. 10 to validate the results?

(c) Why are different peaks observed in absorption and refractive index measurements? It is also observed in dielectric measurements. Justify

(d) Why is DOS very low for positive energy as compared to negative energy? Justify

(d) The conclusion should be very specific about the outcome of the study.

Author Response

We want to thank the reviewer for his / her careful reading of the manuscript and his / her constructive remarks. To improve and clarify the manuscript, we took the comments on board. Please find below a detailed point-by-point reply to all comments.

Reviewer 2

Comments and Suggestions for Authors

In this manuscript, the authors have presented the Electronic Structure and Optical Properties of Inorganic Perovskites using DFT calculations. Following are the comments that need to be addressed:

(a) The introduction part should include the novelty and need of the study in a systematic manner.

Response:

We appreciate the reviewer's feedback. We modified the introduction in the manuscript (page 1, lines 56-58, and lines 68-74).

(b) The authors have calculated the value of the band gap in Fig. 5. Is it possible to calculate the band gap from absorption spectra as shown in Fig. 10 to validate the results?

Response:

Yes, but calculating the band gap from Figure 10 is unnecessary in this study.

(c) Why are different peaks observed in absorption and refractive index measurements? It is also observed in dielectric measurements. Justify

Response:

A complex absorption and dielectric spectrum with multiple peaks can result from the combination of excitonic effects, band-to-band transitions, quantum confinement, phonon modes, defects, and anharmonic effects in CsPbX3 perovskite materials. The specific energies and intensities of these peaks can provide important information about the material's electronic and structural properties, which are important for understanding its behavior and optimizing its performance in various applications. This has been added to pages 10 and 11, lines 218-229.

(d) Why is DOS very low for positive energy as compared to negative energy? Justify

Response:

The difference in the density of states for positive and negative energy levels in CsPbX3 perovskites may be due to the material's specific electronic band structure. In general, perovskite materials have a variety of energy bands, including valence and conduction bands. The electronic band structure governs how energy levels are distributed within these bands and how electrons can move between them.

 (e) The conclusion should be very specific about the outcome of the study.

Response:

The conclusion has been improved.(page 12, lines: 258-261).

Thank you so much

Hamid
